# Postharvest Treatment with Abscisic Acid Alleviates Chilling Injury in Zucchini Fruit by Regulating Phenolic Metabolism and Non-Enzymatic Antioxidant System

**DOI:** 10.3390/antiox12010211

**Published:** 2023-01-16

**Authors:** Alejandro Castro-Cegrí, Sandra Sierra, Laura Hidalgo-Santiago, Adelaida Esteban-Muñoz, Manuel Jamilena, Dolores Garrido, Francisco Palma

**Affiliations:** 1Department of Plant Physiology, Facultad de Ciencias, University of Granada, 18071 Granada, Spain; 2Department of Nutrition and Bromatology, University of Granada, 18071 Granada, Spain; 3Department of Biology and Geology, Agrifood Campus of International Excellence (CeiA3), University of Almería, 04120 Almería, Spain

**Keywords:** Abscisic acid, antioxidants, fruit, zucchini, phenylpropanoid, phenols, chilling, postharvest

## Abstract

Reports show that phytohormone abscisic acid (ABA) is involved in reducing zucchini postharvest chilling injury. During the storage of harvested fruit at low temperatures, chilling injury symptoms were associated with cell damage through the production of reactive oxygen species. In this work, we have studied the importance of different non-enzymatic antioxidants on tolerance to cold stress in zucchini fruit treated with ABA. The application of ABA increases the antioxidant capacity of zucchini fruit during storage through the accumulation of ascorbate, carotenoids and polyphenolic compounds. The quantification of specific phenols was performed by UPLC/MS-MS, observing that exogenous ABA mainly activated the production of flavonoids. The rise in all these non-enzymatic antioxidants due to ABA correlates with a reduction in oxidative stress in treated fruit during cold stress. The results showed that the ABA mainly induces antioxidant metabolism during the first day of exposure to low temperatures, and this response is key to avoiding the occurrence of chilling injury. This work suggests an important protective role of non-enzymatic antioxidants and polyphenolic metabolism in the prevention of chilling injury in zucchini fruit.

## 1. Introduction

Abscisic acid (ABA) is a phytohormone that plays a key role in the growth and development of vegetables and as a regulator of defense against several abiotic stresses, such as salinity, drought, desiccation and low temperatures [1,2,3]. During the postharvest storage of some fruit, such as zucchini and cucumber, low-temperature conservation is a technique widely used with the aim of reducing the metabolic rate to retain the quality of these immature fruit commodities. However, cold stress induces several damages in this subtropical fruit, known as chilling injury (CI). In previous works, this physiological disorder has been studied comprehensively in zucchini fruit (*Cucurbita pepo*), revealing that it is variety dependent [4,5,6], and different physical and chemical treatments that contribute to the improvement of defense against cold stress have been reported, such as preconditioning and exogenous application of polyamines, gamma-aminobutyric acid and nitric oxide [7,8,9,10]. In addition, Carvajal et al. (2017) [11] have reported that the regulation mediated by exogenous ABA is of great importance in this process, mediating the induction of chilling tolerance in zucchini fruit. To unravel the mechanisms underlying this effect, an RNA-seq analysis associated with ABA-induced tolerance has been carried out [12]. However, the role of ABA in the antioxidant defense of zucchini fruit against cold stress remains unknown.

A low temperature increases the production of reactive oxygen species (ROS), inducing oxidative stress when this increment exceeds the capacity of the tissue to scavenge them, leading to cellular damage, and in zucchini fruit, to CI [13,14]. The antioxidant defense mechanisms in the plant cell are composed in part of enzymatic systems, such as superoxide dismutase, catalase and glutathione peroxidase activities, and the Halliwell-Asada pathway [15,16], as well as non-enzymatic antioxidants, such as tocopherol, carotenoids, ascorbate, glutathione and polyphenolic compounds [17,18,19,20]. In zucchini fruit, different treatments, such as preconditioning or the application of putrescine, improve the cold tolerance of this fruit, increasing concomitantly the antioxidant metabolism [7,21]. ABA has also been reported as an important molecule for abiotic stress tolerance in several plant species that enhances the cell antioxidant defense [22,23]. ABA is one of the products of the carotenoid metabolic pathway. In fact, in plant cells, the accumulation of carotenoids after ABA treatments has been observed [24], these natural pigments being important non-enzymatic antioxidants. The application of ABA also increases the antioxidant capacity in fruit through an upregulation of ascorbate and phenylpropanoid pathways [25,26,27,28,29]. In this way, the increase in antioxidant mechanisms mediated by ABA would be responsible for reducing CI in zucchini fruit. The aim of this study was the evaluation of the potential effects of the exogenous application of ABA, just after harvest, on zucchini fruit quality during cold storage, with a focus on the main non-enzymatic antioxidants and a more comprehensive look at the phenolic metabolism.

## 2. Materials and Methods

### 2.1. Sample Material, Treatments and Cold Storage Conditions

Zucchini fruit of the commercial variety, such as Sinatra, a variety that is sensitive to chilling (Clause-Tezier), was supplied by “Hortofrutícola La Ñeca S.L.” (Almeria, Spain). Three replicates were prepared per treatment and storage period, each consisting of 6 fruits of similar size. Immediately after harvest, 18 fruits were sampled to study the fruit characteristics before the application of treatments (T0), and 108 fruits were grouped in 2 lots (ABA treatment and control). The fruit was dipped at 20 °C for 20 min in 0.5 mM of ABA or in distilled water as a control, dried for 2 h, and then stored in a temperature-controlled chamber, in permanent darkness, at 4 °C and 85–90% of RH, for 1, 5 and 14 days. After the storage period, the whole exocarp of each fruit was removed, mixed per replicate, frozen and powdered in liquid nitrogen, a part stored at −80 °C, and the rest was lyophilized.

### 2.2. TEAC, DPPH and FRAP Assays

The trolox equivalent antioxidant capacity (TEAC) in the samples was determined by the ABTS^+^ decolorization assay [30] with some modifications. The results were calculated according to a calibration curve and expressed as mg trolox kg^−1^ fresh weight. The DPPH scavenging radical assay was conducted using the method proposed by Brand-Williams et al. (1995) [31]. The ferric reducing antioxidant power assay (FRAP) was conducted according to Benzie and Strain (1996) [32] with some modifications. The modifications of TEAC and FRAP assays were described previously by our research group [8,21].

### 2.3. Ascorbate Content

Ascorbate was extracted using the method described by Carvajal et al. (2015) [7]. It was then quantified by HPLC in an Agilent 1260 Infinity system equipped with an Agilent ZORBAX Eclipse plus C18 column (150 mm × 4.6 mm id, 3.5 μm) (flow rate: 0.4 mL min^−1^; isocratic conditions: 85% Milli-Q water pH 3 and 15% methanol, run time of 7.5 min; detection: 254 nm). Results were calculated according to a calibration curve and expressed as mg ascorbate per kg of fresh weight.

### 2.4. Carotenoids Quantification

To set up the carotenoids quantification, 100 mg of the lyophilized sample was homogenized with 2 mL hexane, containing the internal standard trans-β-APO-8′-carotenal (0.3 μg/mL), butylated hydroxytoluene (1 mg/mL) and 1.5 mL potassium hydroxide solution in methanol (1 M). The mixture was shaken in darkness for 2 h at room temperature. Subsequently, the carotenoids were extracted and saponified by a saturated sodium chloride solution. This mixture was vortexed for 30 s and centrifuged at 9000× *g* for 5 min. The upper nonpolar hexane phase was collected and evaporated to dryness under a stream of air and then redissolved in 1 mL of isopropanol.

Carotenoids were analyzed by HPLC using a 1260 Infinity Agilent system equipped with a YMC C30 column (250 × 4.6 mm, 3 μm), using mobile phase of acetonitrile/water (95:5) (A) and methyl tert-butyl ether/acetonitrile/water (85:10:5) (B), at a flow rate of 1.2 mL/min. The following gradient was used (min/%A): (0/100%), (10/75%), (23/15%), (25/15%), and (27/100%). The final step was held for 3 min before regenerating the column. Carotenoids were detected using a photodiode array detector (DAD) at 450 nm and identified using the retention times of respective analytical standards.

### 2.5. Polyphenolic Metabolism

#### 2.5.1. Phenylalanine Content

To measure the phenylalanine content, the amino acid extraction was performed as described by Palma et al. (2015) [33] with some modifications. The lyophilized sample was homogenized in 1.5 mL of cold extraction medium (ethanol/choloroform/HCl 0.1 N) (12/5/1; *v*/*v*/*v*) and then supplemented with 75 µL of internal standard (norvaline 500 ppm). The extract was centrifuged for 10 min at 4 °C and 3500× *g*, the supernatant was transferred to a new tube, and then trichloromethane and HCl 0.1 N were added. The upper phase was derivatized with O-phthalaldehyde (OPA). Phenylalanine was analyzed by HPLC (Agilent 1260 Infinity) with an Agilent Poroshell HPH-C18, 4.6 × 150 mm column and a fluorometer using excitation and emission wavelengths of 340 and 450 nm. Phenylalanine was eluted at a flow rate of 1.2 mL min^−1^, with an elution gradient composed by sodium phosphate 10 mM pH = 7.8 (A) and acetonitrile/methanol/water (45/45/10, *v*/*v*/*v*) (B). The gradient profile, expressed as (t [min]; %A), was: (0; 98%), (0.35; 98%), (16.5; 57%), (17; 0%).

#### 2.5.2. Soluble Phenolic Compounds Content

The lyophilized exocarp sample was extracted with methanol 80%, agitated in darkness for 1 h at 4 °C, and centrifuged at 5000× *g* for 10 min at 4 °C. The supernatant obtained was hydrolyzed with 2 M HCl for 1 h at 96 °C in a sealed vial. When the mixture was at room temperature, anhydrous diethyl ether was added, vortexed for 30 s and centrifuged at 5000× *g* for 5 min to extract the soluble phenolic compounds. The upper phase was collected and evaporated to dryness under a stream of air, and then redissolved in methanol 50%. Finally, the phenolic content was determined by the Folin–Ciocalteu method according to Singleton et al. (1999) [34]. The content of phenolic compounds was expressed as g of gallic acid per kg of dry-weight.

#### 2.5.3. Flavonoid Content

To measure flavonoid content, the colorimetric method described by Chang et al. (2002) [35] was used with some modifications. After mixing 100 µL of extract with 100 µL of potassium acetate 1 M, 500 µL of aluminum chloride 10% (*w*/*v*) and 300 µL of distilled water, the mixture was then incubated for 45 min at room temperature in permanent darkness. After, the absorbance was measured at 415 nm. The total flavonoid content was calculated using a calibration curve of quercetin and expressed as mg quercetin per kg of dry weight.

#### 2.5.4. Phenylalanine Ammonia Lyase Activity

The phenylalanine ammonia lyase (PAL) activity was measured according to Civello et al. (1997) [36] with modifications. The exocarp, grounded in liquid nitrogen, was homogenized in 0.1 M sodium phosphate buffer pH = 8.8, containing 2 mM EDTA, 10 mM 2-mercaptoethanol, 0.1% triton X-100 (*v*/*v*) and 5% polyvinylpolypyrrolidone (*w*/*v*). The extracts were shaken at 4 °C for 1 h and centrifuged at 13,000× *g* at 4 °C for 10 min. The reaction mixture was composed of 0.05 M tris-HCl buffer pH = 8.8, phenylalanine 100 mM and supernatant, in proportion (8:1:1; *v*/*v*/*v*). This mixture was incubated for 30 min at room temperature, and then 20% trichloroacetic acid (TCA) (*w*/*v*) was added to stop the reaction. Finally, the samples were centrifuged at 13,000× *g* for 15 min and measured at 290 nm. PAL activity was calculated according to a calibration curve of cinnamic acid and expressed as U/mg protein, where 1 unit was expressed as an increase of 0.001 of absorbance per hour.

#### 2.5.5. Polyphenol Oxidase and Peroxidase Activities

Polyphenol oxidase (PPO; EC 1.14.18.1) and peroxidase (POD; EC 1.11.1.7) activities were determined in the exocarp grounded in liquid nitrogen, homogenizing in 0.1 M sodium phosphate buffer pH 6.5 (1:2, *w*/*v*). Homogenates were centrifuged for 20 min at 4 °C and 20,000× *g*, and supernatant proteins were precipitated with ammonium sulphate at 100% saturation. Precipitated proteins were collected by centrifugation for 15 min at 4 °C and 15,000× g, resuspended in 3 mL of 0.1 M sodium phosphate buffer pH 6.5, and dialyzed at 4 °C in the same buffer. The reaction mixture for PPO activity was composed by 0.1 M sodium phosphate buffer, pH 6, 20 mM pyrocatechol, and 0.1 mL of protein extract. The increase in absorbance at 410 nm was recorded for 1 min, and the activity was calculated according to a calibration curve prepared with catechol and expressed as µg per s and kg of protein. The reaction mixture for POD activity was composed of 0.1 M sodium phosphate buffer pH 6.8, 72 mM guaiacol, 118 mM hydrogen peroxide, and 0.1 mL of protein extract. POD was assayed by measuring absorbance at 470 nm, and the results were expressed as mg H_2_O_2_ per s and kg of protein.

#### 2.5.6. Protein Determination

Total protein concentration in the extracts was measured according to the method proposed by Bradford (1976) [37].

### 2.6. Quantification of Soluble Phenolic Compounds

The extraction was carried out with lyophilized exocarp tissue and methanol 80%, stored in darkness for 1 h at 4 °C, and centrifuged at 5000× *g* for 10 min at 4 °C. The supernatant obtained was mixed with anhydrous diethyl ether, vortexed for 30 s, and centrifuged at 5000× *g* for 5 min. The upper phase was collected and evaporated to dryness under a stream of air, and then redissolved in methanol 50%.

Some of the most notable phenolic compounds against chilling injury cited in the literature were quantified. The sample was injected in an UPLC with an ACQUITY UPLCr HSS T3 1.8μm column at a flow rate of 0.4 mL min^−1^ for 12 min using an elution gradient composed by acetic acid 5% (A) and acetonitrile (B) as mobile phases. The gradient profile, expressed as (t [min]; %A), was: (0; 95%), (5; 50%), (5.1; 0%), (10; 0%), and (10.1; 95%). A Xevo TQ-S triple quadrupole mass spectrometer was used to quantify the soluble phenolic compounds present in the sample, with the transitions shown in Appendix A.

### 2.7. Statistics

The experiment was totally randomized. The statistical analysis was carried out using the SPSS 25.0 program (SPSS Inc., Chicago, IL, USA). Means were compared with Duncan’s least significant differences test (*p* < 0.05).

## 3. Results and Discussion

The importance of the phytohormone ABA in the regulation of cold tolerance in zucchini fruit has been proven by comparing two varieties with different chilling tolerances [11]. However, this work is an attempt to elucidate the antioxidant mechanisms underlying the ABA response to cold stress in fruit throughout postharvest storage. In this experiment, weight loss in the control fruit was higher than the ABA treatment after 5 and 14 days of storage at 4 °C. Moreover, ABA-treated fruit showed significantly less CI than the control fruit during prolonged storage (Table 1). This response confirms the results described in previous work [11].

### 3.1. Evaluation of ABA Effect on Non-Enzymatic Antioxidants

The TEAC, DPPH and FRAP assays, as well as ascorbate content, were measured to determine the antioxidant capacity of zucchini fruit during cold storage, the overall data is presented in Figure 1. Results obtained showed a rise in antioxidant capacity due to ABA application, specifically at the beginning of storage, when all the determinations showed significantly increased rates in fruit treated with ABA. However, in the control fruit, the antioxidant capacity measured by the ABTS, DPPH and FRAP assays either decreased or remained the same during days 1 and 5 of storage with respect to the harvest day (Figure 1).

By contrast, in ABA-treated fruit, a significant increase in these parameters was detected on day 1 of cold storage. This prompt reaction in the fruit treated with ABA during storage could be responsible for the more effective adaptation to cold stress. In a cold-resistant variety of zucchini fruit, such as “Natura,” an increase in ABA content was detected on day 1 of storage, whereas this increase was not measured in the cold-sensitive variety, “Sinatra” [11], which could indicate that the resistance is happening shortly after the fruit is exposed to the cold. The ascorbate content revealed the highest induction due to ABA application, with an increase of around 40% with respect to the control fruit from the first day of cold storage until the end of the experiment (Figure 1). In zucchini fruit, different treatments that improved the chilling tolerance induced the accumulation of ascorbate [7,21]. It has also been reported that the increased content of ascorbate in the fruit of other species is responsible for their resistance to different abiotic stresses. In tomatoes, the overexpression of several genes involved in the increase of this vitamin resulted in fruit that was better adapted to stress conditions [38,39], and in peaches, a hot air treatment that improved their resistance to CI also increased the transcription of genes related to ascorbic acid production [40]. The increase in vitamin C meditated by ABA has been proven to also occur in other fruit, such as blueberries or strawberries [19,25].

Other metabolites with a great antioxidant capacity in fruit are carotenoids [18,41,42]. The mechanism controlling the carotenoid metabolism during postharvest is complex; however, it has been proven that ABA also affects its synthesis and accumulation in tomato fruit [43]. To assess how ABA modulates the content of carotenoids, lutein, zeaxanthin, alfa-carotene and beta-carotene were measured individually (Figure 2).

In the control fruit, the four carotenoids measured decreased the first day of cold storage with respect to the freshly harvested fruit and more intensely in the case of zeaxanthin and beta-carotene. However, the levels of lutein, alpha-carotene and beta-carotene in fruit treated with ABA did not diminish significantly with respect to freshly harvested fruit during postharvest storage, although zeaxanthin was still lower than fruit at harvest and higher than control fruit on days 1 and 5. A study with mandarin fruit revealed a strong negative correlation between the levels of carotenoids in the peel at harvest and the CI index at the end of the postharvest cold storage [44]. Therefore, maintaining carotenoid content during postharvest is key to withstanding cold stress in the fruit, and this response is due to ABA treatment in zucchini fruit. In our study, the most significant change was detected for beta-carotene, with a sharp increase on day 1 of cold storage for ABA-treated fruit and a 30% higher content than what was measured in fruit at harvest (Figure 2). The interaction of ABA and beta-carotene has been reported in transgenic sweet potato plants with a suppressed beta carotene hydrolase. These plants contained higher amounts of beta carotene, and the content in ABA-treated fruits increased, along with enhanced tolerance to abiotic stress [45]. In tomatoes, the overexpression of a lycopene *b*-cyclase in the fruit resulted in increased beta carotene levels, along with an incremental increase in the amount in ABA-treated fruit, and the fruit showed a prolonged shelf life [20]. In our case, the addition of ABA triggered an incremental increase in beta-carotene that could be responsible for a better cold shelf life for zucchini fruit.

#### 3.1.1. Effect of ABA in the Phenylpropanoid Pathway

The phenylpropanoid metabolism produces secondary metabolites in plants, such as flavonoids and phenolic compounds, which are powerful antioxidants that play a key role in plant defense against biotic and abiotic stresses [46,47,48]. The first enzymatic step of phenylpropanoid biosynthesis starts with the deamination of phenylalanine by the enzyme phenylalanine ammonia lyase (PAL) to yield cinnamic acid and downstream phenylpropanoids [49]. Understanding the mechanisms involved in the synthesis and accumulation of these compounds in plants is of vital importance to developing fruit with greater resistance and tolerance to biotic and abiotic stresses [50]. Several studies have reported the effect of abscisic acid as an activator of the phenylpropanoid pathway, thus increasing the number of antioxidants [27,51]. To explore the influence of ABA on the pathway during zucchini fruit postharvest, phenylalanine content and phenylalanine ammonia lyase (PAL) activity were measured, as well as the total soluble phenolic compounds and flavonoids content. An incremental increase in the amount of phenylalanine was detected in the control and treated fruit during cold storage, but this increment was higher during days 1 and 5 when the fruits were treated with ABA (Figure 3). It has been reported that a phenylalanine treatment activates the phenylpropanoid pathway and enhances chilling tolerance in mango fruit due to this treatment inducing an accumulation of flavonoids [52].

The PAL activity, responsible for the start of the pathway, was enhanced by ABA treatment at day 1 respect to control fruit (Figure 3), due to the ability of ABA to alter protein synthesis a few hours after treatment, this rapid response is important in triggering an overall antioxidant mechanism that protects against chilling injury [53]. The phenylpropanoid pathway starts when PAL catalyzes the formation of trans-cinnamic acid through non-oxidative deamination of phenylalanine. Thus, the increase in both the amount of phenylalanine and the enzyme PAL with ABA would lead to an activation of the phenylpropanoid pathway and be responsible for the better adaptation of the fruit to cold stress conditions. It has been reported in several fruits, such as mango, tomato and cucumber, that treatments that increase PAL expression and/or synthesis of phenylalanine also increase phenolic and flavonoid biosynthesis during postharvest [52,54,55]. To check if this increase in phenylalanine and PAL had an effect on the phenylpropanoid pathway in zucchini fruit throughout the cold storage period, soluble phenolics and flavonoids were measured. As expected, the main differences appeared on the first day of cold storage, with an increase in the content of both metabolites, especially in total phenolic compounds, which increased by 32.5% in ABA-treated fruit with respect to the control fruit (Figure 3). This increase in phenolics would protect the fruit against reactive oxygen species, thus improving its defense against cold stress immediately after the stress is detected and also improving its nutritional quality. The relationship between an augment of phenolics and better fruit quality has also been reported in other fruit, such as sweet cherries [56], papaya [57] and oranges [58].

#### 3.1.2. Effect of ABA on Phenol-Oxidizing Enzymes

The rise in the number of phenolics in zucchini fruit during the first day of low-temperature storage due to the activation of its biosynthesis is important in preserving the fruit from cold stress, but this increase could also be due to a reduction in their degradation. The main enzymes involved in the catabolism of phenolic compounds in fruit are polyphenol oxidases (PPO) and peroxidases (POD). PPO and POD activities are important throughout the postharvest period because these enzymes catalyze the browning reactions of phenolic compounds, generally when cellular damage occurs and the substrates are released from the vacuole [59]. Thus, retaining their activity would also help to maintain the nutritional quality of the fruit, reducing the degradation of phenolic compounds, preventing flesh-browning, and extending the storage period during postharvest [60,61,62].

This study proved that cold storage activated the polyphenol oxidase activity in the zucchini control fruit, with a maximum rate of 5 days of storage, causing around a 300% incremental increase with respect to freshly harvested fruit (Figure 4). However, in fruit treated with ABA, the activity of this enzyme remained the same as the measurement in the freshly harvested fruit on days 1 and 14 and showed an important reduction after 5 days of storage with respect to the control fruit. For POD, the main changes happened at the beginning of storage when the control fruit had an incremental increase of around 3-fold, whereas the increase of peroxidase activity in the ABA-treated fruit on day 1 was significantly lower. In pomegranate fruit stored at 4 °C, there was a positive correlation between ion leakage and PPO and POD activities [63]. This loss of electrolytes is associated with damage to the membranes and the appearance of CI.

### 3.2. Evaluation of ABA Effect on Individual Phenolic Compounds

In different fruits, it has been demonstrated that there is a relationship between ABA and the phenylpropanoid pathway, increasing the activity of some key steps [26] and resulting in an accumulation of a wide variety of phenolic and flavonoid compounds [27,64]. Thus, in order to gain a more comprehensive understanding of the implication of this phytohormone in the phenolic metabolism of zucchini fruit, some of the most important hydroxybenzoic acids (vanillic, ellagic), hydroxycinnamic acids (ferulic, *p*-coumaric), and flavonoids (quercetin, isorhamnetin, myricetin, naringenin, luteolin, rhoifolin and chrysoeriol) were quantified (Table 2). Previous experiments conducted in our group have detected these species of phenolics as the most relevant antioxidants in zucchini, and these species also appear in the literature as being the most remarkable.

Vanillic acid is a natural antioxidant of fruits and vegetables [65] that can be added to some foods to improve flavouring and olfactory properties [66]. In this study, vanillic acid was the hydroxybenzoic compound most affected by the ABA treatment, increasing its levels by around 35 and 25 percent after 5 and 14 days of cold storage, respectively, thus maintaining its concentration throughout the postharvest period (Table 2). In mango and olive fruit [67,68], vanillic acid was the main polyphenolic compound measured throughout the postharvest storage because this compound is also added to fruit to prevent decay [69]. The ellagic acid content was augmented during low-temperature storage, however, no significant differences were found for ellagic acid between the control and ABA-treated fruit.

A similar tendency was observed in the case of hydroxycinnamic acids (Table 2). Ferulic acid and *p*-coumaric acid showed a major accumulation after 5 days of storage in ABA-treated fruit compared to control fruit. Ferulic acid was the main hydroxycinnamic acid detected. It is a ubiquitous metabolite in plant tissues that possesses nutraceutical properties [70] and is being used as an additive antioxidant with pharmacological uses for human health [71,72]. It has also been used exogenously in fruit to increase the shelf life [73,74]. With respect to *p*-coumaric acid, it was proven that its effect diminishes the natural decay of fruit during the postharvest period [75]. This is attributed to the activation of the phenylpropanoid pathway by this compound, improving antioxidant capacity and defense against pathogens [76].

Finally, the importance of flavonoids as antioxidant compounds has been tested in several works, protecting fruits and vegetables against oxidative damage throughout the postharvest period [60,77,78,79,80]. Seven classes of flavonoids were quantified in zucchini (Table 2). Quercetin was the main flavonoid detected, reaching levels of about 200 mg/kgDW in fruit at harvest and in fruit treated with ABA. The cold stress caused a significant reduction of quercetin in the control fruit; however, the ABA treatment increased the amount of phenolic compound throughout the entire storage period, maintaining levels similar to freshly harvested fruit (Table 2). Quercetin is used as a treatment with antifungal properties in a wide variety of fruit and vegetables [81,82,83], and its accumulation has been reported as improving chilling tolerances [52].

In peaches, it has been mentioned that quercetin could protect cell membranes from oxidative stress-induced cell death by blocking the cyclooxygenase and lipoxygenase pathways [84]. Quercetin and luteolin are the flavonoid compounds that showed significant differences between the control and ABA-treated fruit throughout the experiments. Luteolin was the flavonoid most affected by the exogenous application of ABA, with its amount increasing between 2- and 3-fold in ABA-treated fruit during cold storage. Luteolin has proven to be an inductor of the phenylpropanoid pathway that contributes to the maintenance of quality in the fruit [85]. Furthermore, it possesses important nutraceutical functions; for example, extracts of plants rich in luteolin have been used in Chinese traditional medicine against various diseases and also have anticancer properties [86]. Isorhamnetin, myricetin, rhoifolin and chrysoeriol, are also of great importance in postharvest maintenance [87,88,89] because they showed important increases due to the ABA treatment, specifically at the beginning of storage (days 1 and 5). Of all the flavonoids studied, only naringenin showed no differences compared to the control fruit.

## 4. Conclusions

Exogenous application of abscisic acid (ABA) increases the chilling tolerance of zucchini fruit during postharvest. The possibility that ABA triggers the defense against CI through an incremental increase of the non-enzymatic antioxidant response has been investigated in this work. For that, ascorbate, carotenoids, and the phenolic metabolism have been analyzed in fruit peel extracts after different periods of cold storage. ABA treatment enhances the chilling tolerance of zucchini through regulation of the antioxidant capacity of the fruit on the first day of cold storage, inducing or maintaining high levels of ascorbate, carotenoids and polyphenols, such as vanillic and ferulic acid or quercetin and luteolin. All of these metabolites also contribute to an increase in the nutraceutical properties of the fruit. These results point to ABA as a phytohormone responsible for the defense against cold stress in zucchini fruit. This knowledge could be used in plant breeding to select for varieties with a higher concentration of ABA to maintain fruit quality during postharvest.

## Figures and Tables

**Figure 1 antioxidants-12-00211-f001:**
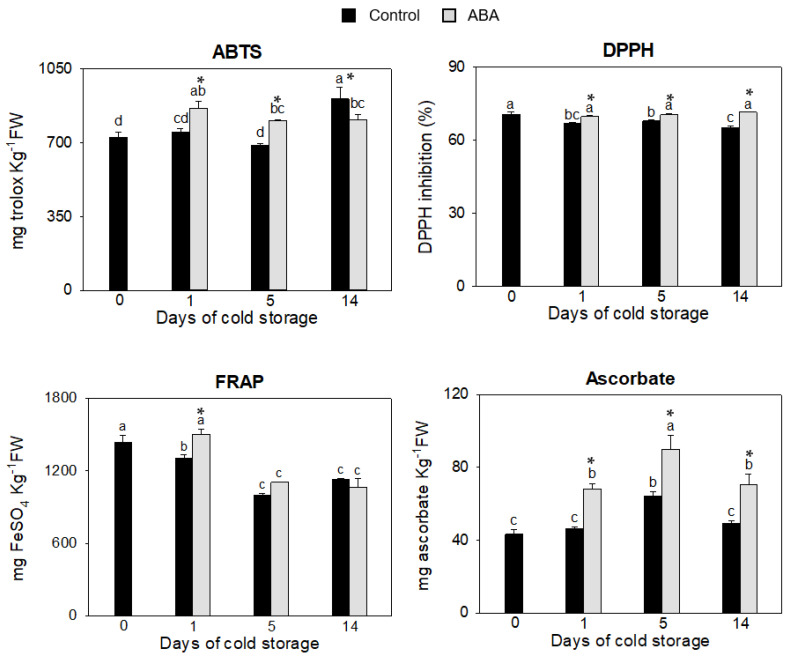
Evaluation of changes in antioxidant capacity by ABTS, DPPH and FRAP assays and ascorbate content in the exocarp of control and ABA-treated fruit at 0, 1, 5 and 14 days of cold storage. Data presented are means ± SE of triplicate samples of six fruit each. Different letters indicate significant differences according to Duncan’s test (*p* < 0.05). The asterisk shows statistically significant differences between treatments for the same storage period.

**Figure 2 antioxidants-12-00211-f002:**
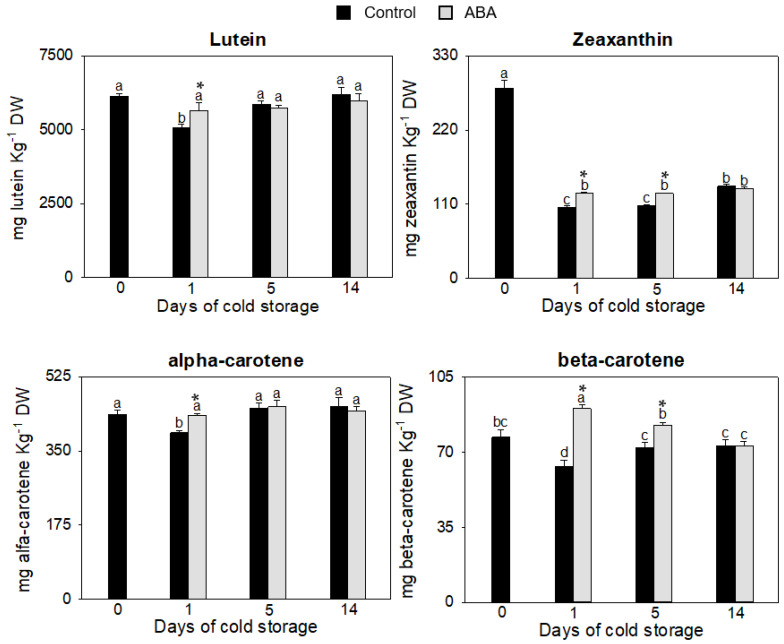
Determination of content of carotenoids, Lutein, Zeaxantin, alfa-carotene and beta-carotene, in the exocarp of control and ABA-treated fruit at 0, 1, 5 and 14 days of cold storage. Data presented are means ± SE of triplicate samples of six fruit each. Different letters indicate significant differences according to Duncan’s test (*p* < 0.05). The asterisk shows statistically significant differences between treatments for the same storage period.

**Figure 3 antioxidants-12-00211-f003:**
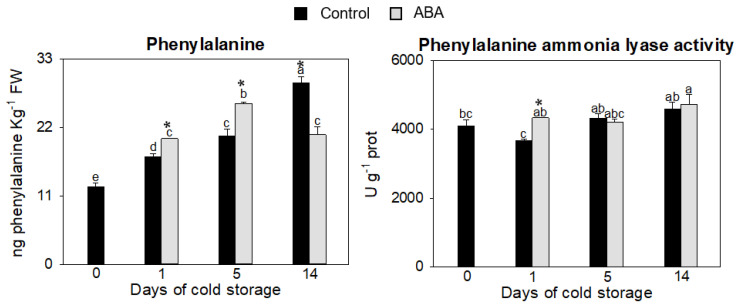
Evaluation of the initiation of phenylpropanoid pathway, by phenylalanine content and phenylalanine ammonia lyase activity, in the exocarp of control and ABA-treated fruit at 0, 1, 5 and 14 days of cold storage. Data presented are means ± SE of triplicate samples of six fruit each. Different letters indicate significant differences according to Duncan’s test (*p* < 0.05). The asterisk shows statistically significant differences between treatments for the same storage period.

**Figure 4 antioxidants-12-00211-f004:**
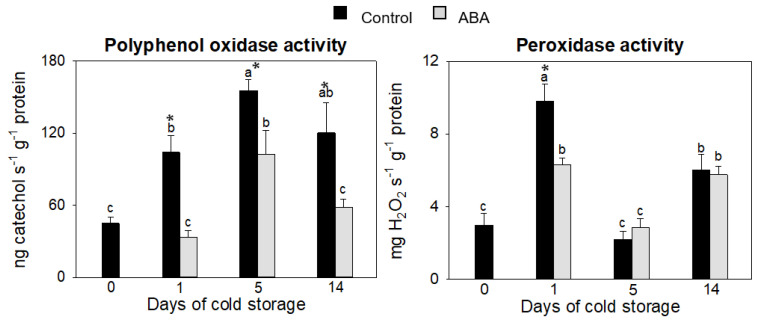
Changes in phenol-oxidizing enzymes peroxidase and polyphenol oxidase of the exocarp of control and ABA-treated fruit at 0, 1, 5 and 14 days of cold storage. Data presented are means ± SE of triplicate samples of six fruit each. Different letters indicate significant differences according to Duncan’s test (*p* < 0.05). The asterisk shows statistically significant differences between treatments for the same storage period.

**Table 1 antioxidants-12-00211-t001:** Percentage of weight loss and chilling injury index during cold storage (1, 5 and 14 days) in control and ABA-treated zucchini fruits. Data presented are means ± SD of triplicate samples of six fruit each. Different letters indicate significant differences according to Duncan’s test (*p* < 0.05). The asterisk shows statistically significant differences between treatments for the same storage period.

Days of Storing at 4 °C	1		5		14	
	Control	ABA	Control	ABA	Control	ABA
Weight loss (%)	1.38 ± 0.1 ^e^	1.48 ± 0.06 ^e^	4.06 ± 0.1 ^c^*	3.37 ± 0.08 ^d^	11.99 ± 0.4 ^a^*	8.39 ± 0.7 ^b^
Chilling Injury	0 ± 0 ^c^	0 ± 0 ^c^	0.18 ± 0.06 ^c^	0.02 ± 0.02 ^c^	2.23 ± 0.2 ^a^*	0.69 ± 0.16 ^b^

**Table 2 antioxidants-12-00211-t002:** Individual phenolic compounds quantified during cold storage (at harvest, 1, 5 and 14 days) in the control and ABA-treated zucchini fruits. Capital letters indicate statistically significant differences (*p* < 0.05) during cold storage in control fruit and lowercase letters for ABA-treated fruit. The asterisk shows statistically significant differences between treatments for the same storage period.

Days of Storing at 4 °C	0	1		5		14	
	At Harvest	Control	ABA	Control	ABA	Control	ABA
Hydroxybenzoic acids							
Vanillic acid (mg/kgDW)	27.3 ± 2.4 ^Aab^	22.8 ± 1.5 ^AB^	22.5 ± 0.6 ^b^	20.4 ± 0.3 ^B^	27.9 ± 1.4 ^a^*	20.7 ± 0.2 ^B^	25.8 ± 1.4 ^ab^*
Ellagic acid (µg/kgDW)	5.7 ± 0.1 ^Bb^	19.8 ± 0.6 ^A^	19.8 ± 1 ^a^	18.6 ± 0.8 ^A^	18.1 ± 0.4 ^a^	18.7 ± 0.5 ^A^	18.3 ± 0.2 ^a^
Hydroxycinnamic acids							
Ferulic acid (mg/kgDW)	8.9 ± 0.4 ^Bb^	3.6 ± 0.5 ^D^	3.3 ± 0.2 ^c^	5.4 ± 0.1 ^C^	8.7 ± 0.5 ^b^*	13.3 ± 0.5 ^A^	12.8 ± 0.7 ^a^
Coumaric acid (µg/kgDW)	21.3 ± 1.4 ^Aa^	25.2 ± 3.4 ^A^	25.8 ± 1 ^a^	ND^B^	13.5 ± 0.8 ^ab^*	ND^B^	ND^B^
Flavonoids							
Quercetin (mg/kgDW)	187.0 ± 30.6 ^Aa^	147.1 ± 7.9 ^AB^	207.7 ± 13.9 ^a^*	181.5 ± 1.8 ^A^	196.8 ± 3.6 ^a^*	130.2 ± 9.3 ^B^	174.4 ± 18.5 ^a^*
Isorhamnetin (µg/kgDW)	30.5 ± 0.4 ^Bc^	31.9 ± 0.7 ^B^	35.6 ± 0.1 ^b^*	34.2 ± 0.1 ^B^	40.1 ± 3.7 ^ab^*	46.5 ± 4.9 ^A^	43.1 ± 1.3 ^a^
Myricetin (mg/kgDW)	2.8 ± 0.1 ^ABb^	2.7 ± 0.1 ^AB^	3.8 ± 0.1 ^a^*	3.0 ± 0.1 ^A^	2.9 ± 0.4 ^b^	2.3 ± 0.3 ^B^	2.6 ± 0.1 ^b^
Naringenin (µg/kgDW)	10.5 ± 1.4 ^Aa^	9.4 ± 0.8 ^A^	10.3 ± 0.8 ^a^	9.1 ± 0.7 ^A^	11.0 ± 0.3 ^a^	9.8 ± 0.8 ^A^	10.8 ± 0.6 ^a^
Luteolin (µg/kgDW)	16.5 ± 0.1 ^ABb^	20.8 ± 2.0 ^A^	43.6 ± 1.2 ^a^*	13.9 ± 1 ^B^	44.8 ± 6 ^a^*	12.6 ± 2.3 ^B^	39.8 ± 6.2 ^a^*
Rhoifolin (µg/kgDW)	16.9 ± 0.4 ^Aa^	8.8 ± 0.4 ^B^	11.5 ± 1.4 ^b^	7.8 ± 0.5 ^B^	11.0 ± 1.1*	8.8 ± 3.2 ^B^	5.6 ± 0.5 ^b^
Chrysoeriol (µg/kgDW)	24.5 ± 0.3 ^Aa^	6.4 ± 0.5 ^B^	10.3 ± 0.2 ^b^*	4.5 ± 1.5 ^B^	5.9 ± 1.4 ^c^	0.3 ± 0.1 ^C^	4.8 ± 1.4 ^c^*

ND: non detected; Data presented are means ± SD of triplicate samples of six fruit each. Different letters indicate significant differences according to Duncan’s test (*p* < 0.05).

## Data Availability

Data is contained within the article and Appendix A.

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
