# Peer review of "Postharvest Treatment with Abscisic Acid Alleviates Chilling Injury in Zucchini Fruit by Regulating Phenolic Metabolism and Non-Enzymatic Antioxidant System"

_antioxidants, 2023, doi:10.3390/antiox12010211_

Round 1
Reviewer 1 Report
the manuscript provides novel and relevant information on the involvement of the antioxidant balance in zucchini in cold post-harvest, following ABA application.
The manuscript is well-designed and the results sound scientifically correct. Some minor points should be amended before publication:
- why the cultivar Sinatra was chosen? is it based on previous findings? the rationale could be reported
- a space should be added between numbers and "°C"
- the choice of analysing only exocarp is related to its main involvement in CI?
- figure 1: statistically significant differences within each time point (treated vs control) could be provided, e.g. by *, ** and *** for p<0,05, 0,01 and 0,001
- L293: cinnamic acid and downstream phenylpropanoids
- standard deviation could be added in table 1
- the rationale for considering the targeted phenolics could be provided
- L419: Seven classes of flavonoids were quantified in zucchini
- is water content changing during shelf life following ABA application
- reference list must be checked
Author Response
Thank you very much for reviewing the MS and for your interesting comments, Herewith we include our responses to these comments:
- why the cultivar Sinatra was chosen? is it based on previous findings? the rationale could be reported
Answer:
The variety Sinatra was chosen because it is one of the most chilling sensitive variety among the different analyzed in previous works. In L69 it has been included that Sinatra is chilling sensitive
- - a space should be added between numbers and "°C"
Answer:
It has been done
- - the choice of analysing only exocarp is related to its main involvement in CI?
Answer:
Exocarp is the tissue that suffers the main damages during cold conservation in zucchini fruit, and has been used to detect physiological changes during storage and with treatments to prevent chilling injuries
- - figure 1: statistically significant differences within each time point (treated vs control) could be provided, e.g. by *, ** and *** for p<0,05, 0,01 and 0,001
Answer: This has been already included in the figures
- - L293: cinnamic acid and downstream phenylpropanoids
Answer: Included (L322-323)
- - standard deviation could be added in table 1
Answer: Already included
- - the rationale for considering the targeted phenolics could be provided
Answer: This paragraph has been included in the MS (L422-424) : “Previous experiments done in our group have detected these species of phenolics as the most relevant antioxidants in zucchini, and these species also appear in the literature as most remarkable”
- - L419: Seven classes of flavonoids were quantified in zucchini
Done, the sentence has been changed
- - is water content changing during shelf life following ABA application
Answer: To clarify this, a new table (Table 1) has been added, in which Chilling injury and weight loss (same as water content change) is shown
- - reference list must be checked
Answer: References have been checked
Reviewer 2 Report
The manuscript antioxidants-2143743 deals with the effect of ABA treatment on alleviating the chilling injury on zucchini postharvest storage. The manuscript is well structured, with a good introduction of the problem, and a M&M organized and fully descriptive.
The manuscript have a major drawback, the authors are not showing any CI data, detais, pictures or a CI index, although:
* The title SAYS "alleviates chilling injury in zucchini fruit",
* In lines 63-64 ,the authors say "The aim of this study was the evaluation of the potential effects of the exogenous application of ABA, just after harvest, ON ZUCCHINI FRUIT QUALITY during cold storage ... ",
* In line 441, one of the authors' conclusion is "Exogenous application of abscisic acid (ABA) increases chilling tolerance of zucchini during postharvest."
But no data for CI or quality are pressented.
Without CI data the manuscript has a different aim, tilte would be "Postharvest treatment with abscisic acid changes phenolic metabolism and non-enzymatic antioxidant system in zucchini fruit" that are the only data presented.
The manuscript should be accepted after providing the CI data.
Author Response
Dear reviewer: Thank you very much for your comments, mainly you tell us that "the authors are not showing any CI data, detais, pictures or a CI index"
You are right with this suggestion and for that a table (Table 1) has been added, with the information about Chilling injury and weight loss in zucchini during cold conservation and with ABA treatment.
We hope that after this change, the MS can be considered for publication in antioxidants
Round 2
Reviewer 2 Report
The manuscript can now be accepted